# Targeting NADPH Oxidase and Integrin α5β1 to Inhibit Neutrophil Extracellular Traps-Mediated Metastasis in Colorectal Cancer

**DOI:** 10.3390/ijms242116001

**Published:** 2023-11-06

**Authors:** Wenyuan Zhu, Siqi Yang, Delan Meng, Qingsong Wang, Jianguo Ji

**Affiliations:** 1State Key Laboratory of Protein and Plant Gene Research, School of Life Sciences, Peking University, Beijing 100871, China; wenyuanzhu@pku.edu.cn (W.Z.); 2101110562@stu.pku.edu.cn (S.Y.); 1901110500@pku.edu.cn (D.M.); 2Department of Biochemistry and Molecular Biology, School of Life Sciences, Peking University, Beijing 100871, China

**Keywords:** colorectal cancer, metastasis, neutrophil extracellular traps, quantitative proteomics, combination therapy

## Abstract

Metastasis leads to a high mortality rate in colorectal cancer (CRC). Increased neutrophil extracellular traps (NETs) formation is one of the main causes of metastasis. However, the mechanism of NETs-mediated metastasis remains unclear and effective treatments are lacking. In this study, we found neutrophils from CRC patients have enhanced NETs formation capacity and increased NETs positively correlate with CRC progression. By quantitative proteomic analysis of clinical samples and cell lines, we found that decreased secreted protein acidic and rich in cysteine (SPARC) results in massive NETs formation and integrin α5β1 is the hub protein of NETs-tumor cell interaction. Mechanistically, SPARC regulates the activation of the nicotinamide adenine dinucleotide phosphate oxidase (NADPH oxidase) pathway by interacting with the receptor for activated C kinase 1 (RACK1). Over-activated NADPH oxidase generates more reactive oxygen species (ROS), leading to the release of NETs. Then, NETs upregulate the expression of integrin α5β1 in tumor cells, which enhances adhesion and activates the downstream signaling pathways to promote proliferation and migration. The combination of NADPH oxidase inhibitor diphenyleneiodonium chloride (DPI) and integrin α5β1 inhibitor ATN-161 (Ac-PHSCN-NH2) effectively suppresses tumor progression in vivo. Our work reveals the mechanistic link between NETs and tumor progression and suggests a combination therapy against NETs-mediated metastasis for CRC.

## 1. Introduction

Colorectal cancer (CRC) is one of the most common cancers with an increasing incidence [1,2] and accounting for approximately 10% of annual cancer-related deaths worldwide [3]. CRC has a propensity for metastasis. Almost a quarter of CRC patients possess metastases at the time of diagnosis, and 20% of the remaining patients are likely to develop distant metastasis [4]. The 5-year survival rate for patients with metastatic CRC decreases from 65% to 14% [5]. Therefore, it is urgent to develop effective therapeutic strategies to suppress tumor progression.

Metastasis is a complex process involving the participation of immune cells, especially neutrophils, which are the most abundant immune cells in the blood [6]. Neutrophil extracellular traps (NETs) are network structures released by neutrophils. They are composed of DNA and hundreds of proteins, such as the citrullinated modification of histone 3 (citH3), myeloperoxidase (MPO), and neutrophil elastase (NE) [7]. NETs can be released via nicotinamide adenine dinucleotide phosphate oxidase (NADPH oxidase)-dependent or non-dependent pathways in response to different stimuli [8]. Different pathways have different regulatory mechanisms. The main processes involved in the formation of NETs are peptidylarginine deiminase IV (PAD4), which promotes nucleosome histone cleavage, and chromatin decondensation by citrullination [9]. With fragmentation of the cell membrane, the decondensed chromatin mixed with granule proteins is released into the extracellular space to form NETs [10]. NETs were originally demonstrated to capture and kill pathogens [11]. As research progressed, their functions have extended to autoimmune disease, thrombus formation and inflammation [12]. Now, NETs are also regarded as a key factor in the acceleration of tumor metastasis. Dissecting the whole process of NETs-mediated metastasis can provide new therapeutic options for colorectal cancer.

The formation of NETs and cancer progression are interrelated. On one hand, tumors promote NETs formation. It has been reported that NETs tend to be increased in many types of cancer [13,14,15,16]. Cytokines [17], proteins [18], RNA [19], and extracellular vesicles [20] secreted by tumor cells have been reported to promote the release of NETs. However, previous studies have focused more on exogenous factors from tumor cells, and it remains unclear whether endogenous regulators exist. The regulatory mechanisms of NETs formation have not been fully elucidated. On the other hand, NETs can accelerate tumor progression and lead to recurrence [21]. DNA present in NETs can interact with coiled-coil domain containing 25 (CCDC25) to promote breast cancer metastasis [22]. Toll-like receptor 4/9-cyclooxygenase-2 (COX2) signaling [14] and interleukin (IL)-1β/epidermal growth factor receptor (EGFR)/extracellular signal-regulated kinase (ERK) signaling in tumor cells can be activated by NETs [23]. NETs can also act as a physical barrier to protect tumor cells from immune system attacks [24]. Therefore, blocking tumor progression accelerated by NETs is a problem that needs to be addressed. Currently, digestion of NETs is the simplest and most studied treatment option. Deoxyribonuclease I (DNase I) [25], anti-cytokine antibody [26], and protein inhibitors [18] have been reported to eliminate NETs. Although eliminating NETs may be one of the methods to inhibit tumor progression, interfering with the interaction between NETs and tumor cells is also of great importance. However, the mechanism by which NETs promote tumor metastasis in CRC remains unclear and there are no combination therapeutic targets to improve the efficiency of NETs-targeted therapy.

In this study, we mapped a proteomic atlas of NETs-mediated tumor progression. We characterized functional changes of neutrophils in CRC and found that secreted protein acidic and rich in cysteine (SPARC) plays a role as a regulator of NADPH oxidase. Downregulated SPARC places neutrophils in a state of NADPH oxidase hyperactivation, which promotes NETs formation. Subsequently, increased NETs accelerate tumor growth and metastasis via the integrin α5β1 downstream pathways. Metastasis can be effectively suppressed using diphenyleneiodonium chloride (DPI) and ATN-161 (Ac-PHSCN-NH2) to inhibit the formation of NETs and prevent their interaction with tumor cells. Our results elucidate the mechanism of NETs-mediated metastasis and provide a potential therapeutic strategy for CRC.

## 2. Results

### 2.1. NETs Formation Ability Is Enhanced in Colorectal Cancer

We isolated neutrophils from the peripheral blood of CRC patients and healthy donors to evaluate NETs formation capability. Fluorescence observation of DNA and the NETs protein marker citH3 demonstrated that neutrophils derived from CRC patients tend to release more NETs (Figure 1A). Under the same stimulatory conditions, fluorescence quantitation of NETs DNA validated that the formation capability of NETs is enhanced in CRC patients. Following the addition of DNase I, NETs were disrupted (Figure 1B). Further, we used dimethyl sulfoxide (DMSO) to induce differentiation of HL60 cells. Differentiated HL60 (dHL60) cells possessing a neutrophilic phenotype, such as changes in cellular karyotype and increased expression of the neutrophil marker complement component 3 receptor 3 subunit (CD11b) (Appendix A). Differentiated HL60 cells were incubated with plasma from CRC patients and healthy donors, which revealed that plasma from CRC patients can stimulate neutrophils to release more NETs (Figure 1C). By immunofluorescence observation of NETs protein markers citH3 and MPO, we found that NETs infiltrate the tumor site widely. Compared to patients with early-stage CRC, patients with metastatic CRC have more infiltration of NETs (Appendix A).

To further confirm the involvement of neutrophils in CRC progression, we used azoxymethane (AOM) and dextran sulfate sodium (DSS) to create a CRC murine model. In this model, mice display phenotypic weight loss, shortened large intestine length, and hyperplastic colonic polyps (Figure 1D, Appendix A). Immunostaining of neutrophil marker lymphocyte antigen 6 complex locus G6D (Ly6G) demonstrated a large number of infiltrated neutrophils in liver metastases, intestinal polyps, and non-polyp intestinal tissue (Figure 1E–G). Similarly, we found that NETs were present at the tumor site, and the infiltration of NETs was higher in the intestines of CRC mice (Figure 1H and Appendix A). Consistent with CRC patients, we observed enhanced NETs formation ability in CRC mice (Figure 1I,J). Immunoblotting results showed that the expression of citH3 and MPO was upregulated in the plasma of CRC. Notably, patients with metastatic CRC had higher expression of citH3 and MPO (Figure 1K, Appendix A). These data suggest that neutrophils from CRC patients tend to release more NETs. NETs were increased in several tissues including plasma, intestine, and tumor in patients with colorectal cancer, suggesting that NETs are extensively involved in tumor progression.

### 2.2. Increased NETs Accelerate Tumor Progression In Vivo

To confirm the impact of NETs on tumor progression, we performed a series of in vivo experiments. A murine model of NETs was constructed using lipopolysaccharide (LPS) to better characterize the phenotype of increased NETs formation in vivo (Figure 2A), which has been shown to be effective [27,28]. LPS resulted in rapid accumulation of neutrophils in the liver and lung and in situ formation of NETs. Isolated neutrophils from LPS-treated mice also experienced significant NETs formation in vitro [14]. Similarly, we observed neutrophil aggregation in the liver (Figure 2B and Appendix A) and elevated plasma expression of NETs markers in LPS-treated mice (Appendix A). After subcutaneous injection of CT26 cells (mouse-derived undifferentiated colon carcinoma cells that share molecular features with aggressive, undifferentiated, refractory human colorectal carcinoma cells [29]), we observed that tumors grew significantly faster in LPS-treated mice (Figure 2C,D). Ki67 expression was elevated at tumor sites and the infiltration of CD8^+^ T cells was reduced inside the tumors (Figure 2E,F). Subsequently, CT26 cells were injected into the mouse spleen to construct a CRC liver metastasis model, from which we observed that the liver metastatic ability of tumor cells was greatly enhanced in LPS-treated mice. Hematoxylin-eosin (HE) staining results showed that the percentage of liver area replaced by tumor increased from 18% to 54% (Figure 2G,H). These results suggest that tumor growth and metastasis are enhanced with the abundant presence of NETs in vivo. Thus, the increment in NETs is positively correlated with tumor progression. Next, we wanted to elucidate the reason for the increased NETs and the manner by which their presence affects tumor cells.

### 2.3. Downregulation of SPARC Promotes NETs Formation

To elucidate the mechanisms underlying the promotion of NETs formation, we isolated neutrophils from the peripheral blood of 25 CRC patients and 8 healthy donors for quantitative proteomic analysis (Figure 3A, Appendix A). In comparison with healthy donors, the expression levels of 40 proteins were significantly changed in CRC patients, of which 20 were upregulated and 20 were downregulated (Figure 3B, Appendix A). Upregulated proteins were associated with neutrophil degranulation, neutrophil activation, and leukocyte-mediated immunological processes. The expression levels of MPO and NADPH oxidase component neutrophil cytosol factor 4 (NCF4), which are closely related to the formation of NETs, were significantly increased (Appendix A). Downregulated proteins were enriched in cell adhesion, apoptosis, and the negative regulation of angiogenesis (Figure 3D). We noticed that SPARC is downregulated (0.63-fold) in CRC neutrophils (Figure 3C). SPARC is a multifunctional glycoprotein that can bind calcium [30] and is known to play diverse biological and functional roles in tumor progression; however, its effect on the formation of NETs remains unclear. Considering the short lifespan of primary neutrophils(~10–18 h once released in the bloodstream [31]), we used HL60 cells, which can be induced to differentiate into neutrophils, to perform in vitro experiments such as gene editing and NETs quantification. First, we generated SPARC-knockdown HL60 cells (Figure 3E). We then used DMSO to induce their differentiation into dHL60 cells and assayed their NETs formation capacity by fluorescence observation and quantification. In the absence of exogenous stimuli, the release of NETs increased about 1.6-fold after SPARC knockdown (Figure 3F and Appendix A). Fluorescence observation results also showed that NETs formation was enhanced after knockdown of SPARC (Figure 3G).

Studies have shown that exogenous factors represented by cytokines can stimulate neutrophils to release NETs [17]. Our previous data also showed that CRC plasma can promote NETs formation. To explore whether the downregulation of SPARC is caused by exogenous factors, we quantified 13 cytokines in plasma by flow cytometry and found that the overall level of interleukin-6 (IL-6) was significantly increased in CRC patients (Appendix A). In both the AOM/DSS-induced CRC murine model and the LPS-treated murine model, we also observed increased expression of IL-6 (Appendix A). Then, we stimulated dHL60 cells with IL-6 in vitro and found that IL-6 can promote NETs formation through the NAPDH oxidase-dependent pathway (Appendix A). Quantitative proteomic analysis demonstrated that the expression levels of reactive oxygen species (ROS) response-related proteins and apoptotic process-related proteins are significantly increased after IL-6 treatment (Appendix A). Notably, serine/threonine-protein kinase PAK 1 (PAK1) and receptor-interacting serine/threonine-protein kinase 1 (RIPK1) have been reported to be related to NETs formation [32,33]. However, the expression level of SPARC did not change significantly (Appendix A). Moreover, we used culture medium of CRC cell line HCT116 to incubate HL60 cells, and again, SPARC was not changed (Appendix A). These results suggest that neutrophils from CRC patients have characteristic protein expression changes and further affect their function. Downregulation of SPARC is an independent endogenous factor in neutrophils that promotes NETs formation.

### 2.4. SPARC Regulates NADPH Oxidase by Interacting with RACK1

To uncover the NETs-promoting mechanism, we performed quantitative proteomic analysis of SPARC-knockdown HL60 cells. Bioinformatic analysis results showed that significantly upregulated protein functions are enriched in the process of NADPH oxidase activation (Figure 4A and Appendix A). The expression levels of NADPH oxidase process-related proteins are significantly increased (Figure 4B). NADPH oxidase is an essential part of the NETs formation pathway. The activation of NADPH oxidase typically requires the activation of protein kinase C (PKC) or the ERK/mitogen-activated extracellular signal-regulated kinase (MEK) pathway followed by the phosphorylation of intracellular NADPH oxidase component proteins, which complete the assembly of NADPH oxidase [34]. Immunoblotting analysis showed that knockdown of SPARC increases the expression and phosphorylation levels of intracellular PKC and NADPH oxidase intergroup proteins, in addition to the phosphorylation levels of members of the upstream MEK/ERK pathway (Figure 4C and Appendix A). Activation of NADPH oxidase is often accompanied by the generation of ROS. We found that the generation of intracellular ROS is enhanced following SPARC knockdown with or without exogenous phorbol 12-myristate 13-acetate (PMA) stimulation (Appendix A and Figure 4D). Increased ROS initiates NETs formation processes. Among them, peptidylarginine deiminase 4 (PAD4) plays a key role in the NETs formation process. It mediates histone citrullination, which leads to chromatin de-concentration [35]. In SPARC-knockdown HL60 cells, the expression level of PAD4 is elevated (Appendix A).

To further explore the regulatory mechanism of SPARC, we used immunoprecipitation-mass spectrometry (IP-MS) to identify proteins with which SPARC interacts. We performed a co-IP assay using the pCMV-SPARC-3 × Flag plasmid (Figure 4E and Appendix A). Tandem mass spectrometry data showed that the receptor of activated protein C kinase 1 (RACK1) may interact with SPARC. Peptides derived from RACK1 were detected in the experimental group but not in the control group (Appendix A). Through exogenous and endogenous co-IP experiments, we verified that there is an interaction between SPARC and RACK1 under both SPARC overexpression conditions in HEK293T cells and normal physiological conditions in HL60 cells (Figure 4F,G). In addition, immunofluorescence confocal microscopy revealed colocalization of RACK1 and SPARC (Figure 4H). RACK1 is an adaptor protein that mediates PKC phosphorylation [36], while PKC is one of the most important sources of NADPH oxidase activation. These results suggest that SPARC may interact with RACK1 to regulate the activation level of NADPH oxidase (Appendix A). Over-activated NADPH oxidase will increase intracellular ROS generation and promote the formation of NETs.

### 2.5. NETs Promote CRC Cell Proliferation, Migration and Invasion In Vitro

To elucidate the manner in which NETs’ presence affects tumor cells, we focused on the effect of NETs on CRC cells in vitro. We selected three CRC cell lines with various genomic mutations and microsatellite instability (MSI) statuses for in vitro experiments. LoVo and HCT116 cells have the hypermutator phenotype (MSI) while SW620 cells do not. NETs were isolated from PMA-stimulated dHL60 cells and incubated with CRC cells. We found that NETs promote cancer cell proliferation (Figure 5A,B). The transwell chamber assay and wound healing assay demonstrated that the capabilities of tumor cells to migrate and invade are enhanced in the presence of NETs (Appendix A and Figure 5D–G). Following the disruption of NETs using DNase Ι, the enhanced migration, invasion, and proliferation capabilities of CRC cells were abolished. The expression levels of epithelial-mesenchymal transition (EMT) pathway-related proteins were increased following incubation with NETs (Figure 5C and Appendix A). Quantitative real-time polymerase chain reaction (RT-qPCR) results also showed that the mRNA level of E-cadherin decreased and N-cadherin, snail, vimentin, twist, and matrix metalloproteinase 2 (MMP2) increased with NETs incubation (Appendix A). Then, we treated HCT116 cells with NETs for 48 h in vitro and subsequently injected them subcutaneously into nude mice. Tumor cells incubated with NETs were found to grow faster in vivo than controls (Figure 5H,I). Static adhesion experiments showed that NETs have the ability to physically adhere to tumor cells (Appendix A). These results suggest that the presence of NETs can effectively capture tumor cells and enhance proliferation, migration, and invasion capabilities of tumor cells.

### 2.6. NETs Affect CRC Cells through Integrin α5β1 Signaling Pathways

To uncover the effect of NETs on tumor cells, quantitative proteomics was used to analyze functional changes initiated by NETs (Figure 6A, Appendix A). Pathways related to cell proliferation and migration in CRC cell lines were significantly changed after incubation with NETs (Appendix A). Among the proteomic results for HCT116, SW620, and LoVo cells, there were 25 common significantly differentially expressed proteins. It is worth noting that the expression levels of integrin alpha-5 (ITGA5), integrin beta-1 (ITGB1), and ras homolog family member A (RHOA) were significantly increased under the stimulation of NETs (Figure 6B). Integrins are obligate heterodimers composed of alpha and beta subunits; ITGA5 (α-subunit) and ITGB1 (β-subunit) together form integrin α5β1, which mediates a variety of biological functions at the plasma membrane [37].

Accordingly, we wanted to determine whether NETs affect tumor cells through integrin α5β1 signaling. Western blotting results showed that incubation of tumor cells with NETs increased the phosphorylation level of focal adhesion kinase 1 (FAK) and proto-oncogene tyrosine-protein kinase Src (SRC), in addition to the expression of their downstream pathway proteins, which are related to cell migration. The phosphorylation level of Akt was also elevated, as was the expression of Ki67. However, knockdown of ITGB1 attenuated the activation of these signaling pathways (Figure 6C and Appendix A). Moreover, activation of the integrin α5β1 pathway was enhanced as the NETs incubation time and concentration increased (Appendix A). The integrin family comprises 24 transmembrane receptors [38], and according to the specific characteristics of each integrin, different inhibitors are currently being investigated in clinical trials. Here, NETs-incubated HCT116 cells were treated with ATN-161, a small peptide antagonist of integrin α5β1, which attenuated their activation by NETs while not exhibiting a significant cell killing effect (Figure 6D and Appendix A).

NETs may interact with integrins on the surface of tumor cells to mediate their adhesion [39]. We confirmed these results by knocking down ITGA5 or ITGB1, following which the ability of NETs to adhere to tumor cells was greatly reduced (Figure 6E–G). Stable isotopic metabolic labeling (SILAC) was used to further determine the proteins present in NETs that interact with integrin α5β1 on HCT116 cells and proteins containing ^13^C_6_ modification of lysine or arginine were detected by immunoprecipitation-mass spectrometry (IP-MS) (Appendix A). SDS-PAGE and peptides identified by MS demonstrated that ITGB1 was enriched in the experimental group (Appendix A). By screening the only peptides with modifications in the experimental group, we found that interacting proteins with ITGB1 in NETs were mainly enriched in functions such as integrin-mediated cell adhesion (Appendix A). Taken together, these results suggest that NETs can increase the expression level of integrin α5β1 on the surface of tumor cells to enhance NETs-tumor cell adhesion. In addition, NETs also promoted tumor cell proliferation and migration by activating the integrin α5β1 downstream signaling pathways.

### 2.7. Combination Therapy with the NADPH Inhibitor DPI and the Integrin α5β1 Inhibitor ATN-161 Suppresses Tumor Progression

Based on the above observations, we found that downregulated SPARC leads to the overactivation of NADPH oxidase and integrin α5β1 play an important role in the interaction between NETs and CRC cells. Therefore, we further explored the effect of inhibiting NADPH oxidase and integrin α5β1 on tumor progression (Figure 7A). First, we constructed a murine model of CRC with increased NETs via intraperitoneal injection of LPS and subcutaneous injection of CT26 cells in BALB/c mice. Mice were treated with the NADPH oxidase inhibitor DPI and the integrin α5β1 inhibitor ATN-161. The NETs-eliminating agent DNase I was used as a positive control. The body weight of mice in the drug treatment group did not decrease significantly, indicating minimal toxic side effects (Figure 7B). Among the monotherapy groups, DPI had the best therapeutic effect; followed by ATN-161, which blocks the effect of NETs on tumor cells. A combination of DPI and ATN-161 more effectively inhibited tumor growth (Figure 7C,D). Following drug treatment, the infiltration of NETs within the tumors was significantly reduced and that of CD4^+^ T and CD8^+^ T cells was increased (Figure 7E,F).

Then, we constructed a CRC liver metastasis model via intraperitoneal injection of LPS and splenic injection of CT26 cells in BALB/c mice. The results showed that drug-treated mice had significantly fewer tumor metastases and less infiltration of NETs in the liver. The average area of liver replacement of tumor metastases in the control group was 61%, which decreased to 7% and 8% after ATN-161 or DPI monotherapy. After the combination treatment, the liver replacement area decreased to 2%. The combination of DPI and ATN-161 showed the best metastasis-suppressing effect (Figure 8A–D). Moreover, we detected the expression levels of plasma proteins in different drug treatment groups and found that NETs-related proteins are good indicators of disease severity. The expression levels of MPO, citH3, and IL-6 were decreased following drug treatment (Figure 8E,F and Appendix A). These results suggest that simultaneous targeting of NADPH oxidase and integrin α5β1 can effectively reduce NETs formation and inhibit tumor progression with better therapeutic efficacy than monotherapy. Therefore, combination therapy has potential for clinical application. Disease progression can also be indicated by detecting the expression levels of plasma NETs-related proteins.

## 3. Discussion

NETs are extracellular structures released by neutrophils, which have been reported to be increased in cancer patients and promote metastasis [12]. Therefore, the reason behind the increase in NETs has attracted significant attention. Numerous studies have demonstrated that the activation of neutrophils by tumor cells and their microenvironments can lead to the release of NETs [40]. However, fewer studies have comprehensively characterized the internal changes in neutrophils. With the use of quantitative proteomics, we produced a protein atlas and revealed functional changes in neutrophils from CRC patients, which are mainly related to the activation of neutrophils and NADPH oxidase. The neutrophil activation marker arginase-1 (ARG1) [41] and angiogenic factor matrix metalloproteinase-9 (MMP9) were highly expressed, while the expression of C-C motif chemokine 5 (CCL5) and perforin-1 (PRF1) was downregulated. Our data suggests that neutrophils in the blood are highly similar to N2 neutrophils (tumor-promoting neutrophils) in the tumor microenvironment [42]. Through in vitro experiments, we found that the overactivation of NADPH oxidase was caused by the downregulated SPARC. The expression of SPARC is not affected by secretions from tumor cells. This suggests that it is not only factors exerted by tumor cells that can stimulate neutrophils to release more NETs, but neutrophils in CRC patients are also altered to a state that favors tumor progression.

There are two main pathways leading to NETs formation: the NADPH oxidase-dependent pathway and the NADPH oxidase-independent pathway. In the NADPH oxidase-dependent pathway, PKC or the rapidly accelerated fibrosarcoma (Raf)-MEK-ERK pathway is activated [43], which results in the activation of NADPH oxidase and the generation of ROS. Elevated ROS promotes degranulation and the translocation of MPO and NE to the nucleus, promoting histones to drive chromatin depolymerization [44]. The NADPH oxidase-independent process does not require NADPH oxidase and ROS generation during the formation of NETs. In this process, Ca^2+^ is transferred into neutrophils through small conductance potassium (SK) channel member SK3, which can activate PAD4, resulting in citH3 and chromatin de-condensation [45]. Our results demonstrate that the SPARC can regulate the activation of PKC by interacting with RACK1, which increases the expression and phosphorylation of the NADPH oxidase pathway, ultimately leading to elevated intracellular ROS levels. Briefly, downregulation of SPARC resulted in activation of the NADPH oxidase-dependent pathway for NETs formation. SPARC is a matricellular molecule possessing various functions. Conflicting results have been reported regarding the role of SPARC in tumor progression [46]. With respect to immune cells, reports have demonstrated that in lymphoma, the absence of SPARC promotes aberrant interactions between NETs and CD5^+^ B cells [47]. SPARC has also been shown to suppress the function of myeloid-derived suppressor cells (MDSCs) [48]. Here, we found that SPARC is an upstream negative regulator of NADPH oxidase in neutrophils. Although the detailed mechanism by which SPARC interacts with RACK1 is yet to be defined, these findings will help us understand the novel role of SPARC. 

Considering the tumor-promoting effects of NETs, many studies have focused on their elimination. There are three ways to reduce the formation of NETs: digesting NETs, blocking the effect of tumor cells on neutrophils, and inhibiting key proteins in the NETs formation process. The use of DNase Ι is an easy way to degrade NETs. But, systemic biodistribution of DNase I may compromise host defenses against infection [49]. Neutralizing antibodies against NETs stimulating factors (IL-17 [50], IL-8 [17], CXCR1 [24], CXCR2 [51], FcγRIIA [52], etc.) have been shown to be therapeutically effective. Inhibition of NETs formation may enhance the effectiveness of NETs targeting therapies. In this work, we found that both the exogenous factor IL-6 and the endogenous factor SPARC promote the formation of NETs through the NADPH oxidase-related pathway. Although they are independent factors with different functional mechanisms, this also suggests that NADPH oxidase plays a pivotal role in the process of NETs formation in CRC. Thus, NADPH oxidase is a potential therapeutic option for a NETs targeting therapy. DPI is a well-acknowledged NADPH oxidase inhibitor that has been used to dampen a variety of inflammatory responses [53]. In our data, DPI demonstrated a favorable tumor suppression effect, as it effectively inhibited the overactivation of NADPH oxidase in neutrophils, and also possessed tumor-killing ability. It remains unclear whether there exists other toxic side effects related to the prolonged use of DPI. Encouragingly, another study has shown that an ultralow dose of DPI can safely suppress colitis-associated CRC [54]; therefore, DPI-like NADPH oxidase inhibitors have the potential to suppress NETs formation and tumor progression in CRC. In addition to NADPH oxidase, research has been devoted to finding new therapeutic targets, such as PKC [55], PAD4 [56], NE [57], gasdermin D [58], etc. NETs formation is a complex process and the factors that play a dominant role in different diseases vary. Therefore, target selection and evaluation of the therapeutic efficacy of NETs-targeted therapies also need to be further explored.

Targeting NETs alone is not a sufficient anticancer therapy, and multi-target combinations may have better therapeutic effects. Thus, the impact of NETs on tumor cells is noteworthy. Our study demonstrated that integrin α5β1 is a hub protein in NETs-accelerated tumor progression. The integrin α5β1 inhibitor ATN-161 does not affect tumor cell proliferation in vitro, but it inhibits tumor growth in LPS-treated mice, suggesting that cutting off the effects of NETs on tumor cells inhibits tumor progression. A combination therapy that targets NETs formation and blocks its interaction with tumor cells significantly inhibited tumor progression and has a better therapeutic effect than monotherapy. Studies have reported the therapeutic potential of ATN-161 in combination with chemotherapeutic agents [59]. It is unclear whether the effects of NETs on tumor cells are exerted through the whole integrin family or are limited to integrin α5β1, as we also noted changes in the expression of other integrins in tumor cells after incubation with NETs (Appendix A). Other studies have also demonstrated the potential of blocking the effects of NETs on tumor cells to inhibit tumor progression. NETs could exert pro-tumorigenic effects by activating TLR9 and its downstream signaling. In glioblastoma cells, NETs-derived high mobility group protein 1 (HMGB1) bound to receptor for advanced glycation endproducts (RAGE) and upregulated the NF-κB signaling in tumor cells [60]. In hepatocellular carcinoma (HCC), the activation of TLR4 and TLR9 by NETs led to upregulation of cyclooxygenase-2 (COX2), which enhanced the invasiveness of tumor cells. A combined strategy of DNase I plus hydroxychloroquine (HCQ)/aspirin against NETs effectively impaired HCC metastasis [14]. Additionally, we observed decreased T cell infiltration within tumors in mice possessing increased NETs. After drug treatment, T cell infiltration within the tumor was increased, suggesting that increased NETs formation may suppress T cell function. Recently, studies have reported that NETs promote T cell exhaustion [61] and affect Treg cell differentiation [62]. In CRC, inhibition of NETs with DNase I results in the reversal of anti-PD-1 blockade resistance [63]; thus, exploring the relationship between NETs and adaptive immunity may provide novel ideas for tumor therapy. Multi-target combination therapy is the current trend in tumor treatment; however, further research is needed to study drug selection and dose optimization.

In conclusion, our study describes the functional changes in neutrophils from CRC patients and comprehensively elucidates the entire process by which increased NETs accelerate tumor progression. SPARC plays a role as an upstream negative regulatory protein of NADPH oxidase. The downregulation of SPARC places neutrophils in a state of NADPH oxidase hyperactivation, leading to elevated intracellular ROS, which promotes NETs formation. NETs upregulate the expression level of integrin α5β1 in tumor cells to enhance NETs-tumor cell adhesion and promote tumor proliferation and migration. Simultaneous targeting of NADPH oxidase in neutrophils and integrin α5β1 in cancer cells can suppress tumor progression and improves the clinical benefit of NETs-targeted therapy. Therefore, our findings provide a new option for CRC treatment.

## 4. Materials and Methods

### 4.1. Clinical Samples

Peripheral blood samples were obtained from healthy donors and colorectal cancer patients at Beijing Chao-Yang Hospital. Tumor grades were determined by clinicians based on histological features. A total of 33 samples were used for neutrophil proteomics analysis, 23 samples were used for plasma cytokine quantitation, and 24 samples were used for plasma protein immunoblotting. The present study was reviewed and approved by the Chao-Yang Hospital Medical Ethics Committee. All patients were aware of the scientific use of peripheral blood and gave written informed consent. Pathological information regarding CRC patients is described in Appendix A.

### 4.2. Plasma and Neutrophil Isolation

Peripheral blood was centrifuged at 3000 rpm for 10 min. Plasma was stored at −80 °C until further use and peripheral blood mononuclear cells (PBMCs) were isolated using Ficoll-Paque Plus (Cytiva, Marlborough, MA, USA). Immunomagnetic beads (MACS, Miltenyi Biotec, Bergisch Gladbach, Germany) were used to sort CD15^+^ neutrophils. The purity of sorted cells was measured by flow cytometry (FACSVerse, BD, San Jose, CA, USA) using anti-CD11b and anti-CD15 antibody staining (Biolegend, San Diego, CA, USA). For murine neutrophil isolation, Ficoll-Paque Plus and dextran T-500 (Solarbio, Beijing, China) were used. The purity of sorted cells was measured by flow cytometry (FACSVerse) using anti-Ly6G antibody staining (Biolegend).

### 4.3. Cell Culture and Differentiation

HEK293T, HCT116, SW620, and LoVo cells were cultured in DMEM (Hyclone Sydney, Australia) supplemented with 10% fetal bovine serum (Hyclone). HL60 and CT26 cells were obtained from the Cell Resource Center, Institute of Basic Medicine, Chinese Academy of Medical Sciences and cultured in RPMI-1640 (Gibco, Carlsbad, CA, USA) supplemented with 10% fetal bovine serum (Hyclone). HL60 cells were induced to differentiate using RPMI-1640 medium containing 1.25% DMSO (Sigma-Aldrich, Saint Louis, MO, USA) for 4 days. Completion of differentiation was verified by CD11b expression and changes in nuclear morphology. The expression of CD11b was detected by flow cytometry and PCR, and nuclear morphology was observed using a 120 kV transmission electron microscope (FEI Tecnai G2 Spirit).

### 4.4. Animal Models

Animal studies were approved by the IACUC of the Center for Experimental Animal Research (Beijing, China) and Peking University Laboratory Animal Center (Beijing, China, IACUC No. LSC-JiJG-8). Male BALB/c, C57BL/6, and BALB/c-nude mice (6–8 weeks old) were purchased from Beijing Vital River Laboratory Animal Technology Co., Ltd. (Beijing, China).

For the AOM/DSS-induced CRC model, 8-week-old male C57BL/6 mice were selected and injected intraperitoneally with azoxymethane (AOM, 10 mg/kg, Sigma-Aldrich). The control group was injected with saline. Mice were administered regular drinking water for 1 week, after which they were subjected to 3 cycles of dextran sulfate sodium (DSS, 2%, *w*/*v*, MW, 36,000–50,000, MP Biomedicals, Solon, OH, USA) administration for 1 week, followed by a 1-week recovery period. The mice were euthanized on day 60. The abdominal cavity was opened along the abdominal midline and the intestinal segment was cut from the ileocecal part to the rectum, photographed, and the number of tumors was recorded before further analysis.

For the subcutaneous tumor model, HCT116 cells were incubated with or without NETs for 48 h in vitro. A total of 2 × 10^6^ HCT116 cells were injected subcutaneously into 8-week-old BALB/c-nude mice. For the NETs mice subcutaneous tumor model, 8-week-old male BALB/c mice were selected. Mice were intraperitoneally injected with 10 μg lipopolysaccharide (LPS, Sigma), and the control group was intraperitoneally injected with saline. At 24 h after LPS injection, each mouse was injected subcutaneously with 2 × 10^6^ CT26 cells. When the tumor was visible, the length and width were recorded every two days and a tumor growth curve was constructed. Tumor volume calculation formula: tumor volume (mm^3^) = 1/2 × length (mm) × width^2^ (mm^2^). Mice were sacrificed before the tumor volume exceeded 1000 mm^3^.

For the CRC liver metastasis model, 8-week-old male BALB/c mice were selected. Mice were intraperitoneally injected with 10 μg LPS (Sigma-Aldrich), and the control group was intraperitoneally injected with saline. At 24 h after LPS injection, 2 × 10^6^ cells CT26 cells were injected into the spleen. During the operation, the surface of the spleen was kept moist by dropping saline. After 20 days, mice were euthanized, and the intrahepatic metastatic burden was assessed.

For in vivo drug treatment experiments, 2 × 10^6^ CT26 cells were injected subcutaneously or into the spleen of 8-week-old male BALB/c mice after 10 μg LPS (Sigma-Aldrich) injection. Mice were randomly assigned to five treatment groups (8 mice per group): (A) Control (saline); (B) DNase Ι (200 U, Sigma-Aldrich); (C) ATN-161 (1 mg/kg, Selleck, Houston, TX, USA); (D) DPI (3 mg/kg, Selleck); and (E) ATN-161 + DPI. Treatment was started on day 4 after CT26 cell injection and was administered every two days. The length and width of the tumor were recorded every two days and a tumor growth curve was constructed. 

All mouse tissues were fixed by tracheal perfusion with 4% paraformaldehyde (PFA, Solarbio) and embedded in paraffin for subsequent HE staining, immunohistochemistry, and immunofluorescence observation.

### 4.5. Sample Preparation for Quantitative Proteomics Analysis

Protein samples were precipitated with acetone at −20 °C overnight, resuspended in 8 M urea, and sonicated at 4 °C (Bioruptor, Seraing, Belgium). Alkylation was performed using dithiothreitol (DTT) and iodoacetamide (IAA). Samples were digested using LysC (1:100, *w*/*w*, Wako, Osaka, Japan) and trypsin (1:50, *w*/*w*, Promega, Madison, WI, USA) overnight at 37 °C with rotation at 200 rpm. The mixtures were acidified and desalted using Empore C18 StageTips (CDS, Oxford, PA, USA), and the peptides were drained using a refrigerated vacuum centrifuge. Peptide labeling was performed according to the Tandem Mass Tag Labeling Kit (Thermo Fisher Scientific, Waltham, MA, USA). After labeling, peptides were frozen and vacuum-dried and resuspended in ddH2O (pH = 10). Peptides were eluted with a 10–50% acetonitrile gradient, and samples were subsequently frozen and vacuum-dried.

### 4.6. Liquid Chromatography Coupled to Tandem Mass Spectrometry (LC-MS/MS)

Peptides were dissolved in 10 μL 0.2% formic acid and separated on a C18 column by EASY-nLC 1200 nanoliter liquid chromatography (Thermo Fisher Scientific). Chromatographic separation was performed using a linear gradient from 6% to 90% acetonitrile with 0.1% formic acid at a flow rate of 300 nL/min and a gradient time of 194 min. Mass spectrometry data were collected using an Orbitrap Fusion Lumos Tribrid mass spectrometer (Thermo Fisher Scientific). The MS1 was detected in the Orbitrap with a resolution of 120,000. Secondary mass spectrometry MS/MS selected 15 precursor ions with the strongest signal for HCD high-energy collision dissociation (collision energy 37%), and MS2 was detected in the Orbitrap with a resolution of 50,000.

### 4.7. LC-MS/MS Data Analysis 

Raw mass spectrometry data were analyzed using Proteome Discoverer (Version 2.2.0.388, Thermo Fisher Scientific), and MS/MS spectra were searched against the human Uniprot FASTA database (February 2019 edition, 95,556 entries) using SEQUEST-HT (Thermo Fisher Scientific, Waltham, MA, USA). The enzyme was set to trypsin, and the enzyme specificity of trypsin was set to a maximum of 2 deleted cleavages and a minimum peptide length of 6 amino acids. Static modifications were set to carbamoyl methylation of cysteine (+57.021) and TMTpro (+229.163) of lysine residues and peptide N termini. Variable modifications were set to acetylation of N-termini (+42.011) and oxidation of methionine (+15.995). Total precursor ion mass tolerance was set to 10 ppm and the product ion mass tolerance was set to 0.02 Da. A 1% false discovery rate (FDR) was applied at the peptide and protein levels. Peptides were normalized to the total peptide amount. Other parameters followed the default settings.

### 4.8. Bioinformatics Analysis

This study filtered out protein entries with missing values and used proteins with quantitative data in all samples for subsequent bioinformatics analysis. For the TMT experiment with ten channels, 131 channels of each batch were the same mix of samples for data normalization. The screened proteomics data fit a negative binomial distribution and were statistically analyzed using the R package DESeq2 [64] (version 1.24.0). Proteins showing a statistically significant change were selected using the criteria of an adjusted *p*-value < 0.05 and an absolute value of log_2_ fold change > 0.5. Metascape [65] and R package clusterProfiler [66] (version 3.12.0) were used for Gene Ontology (GO) annotation and Kyoto Encyclopedia of Genes and Genomes (KEGG) pathway analysis, respectively. The STRING database (http://string-db.org (accessed on 18 May 2021)) [67] was applied to identify protein networks, and only interactions with experimental evidence and a score greater than 0.4 were selected for this study. The protein interactions were visualized using Cytoscape [68] (version 3.7.2). Volcano plots and boxplots were constructed using the R package ggplot2 (version 3.3.2), and heatmaps were created using the R package pheatmap (version 1.0.12).

### 4.9. Plasma Cytokine Quantitation

Levels of plasma cytokines were detected using the Human Essential Immune Response Panel of the LEGENDplex™ Multi-Analyte Flow Assay Kit (Biolegend), which included interleukin(IL)-4, IL-2, C-X-C motif chemokine ligand (CXCL)-10 (IP-10), IL-1β, tumor necrosis factor (TNF)-α, C-C motif chemokine ligand (CCL)-2 (MCP-1), IL-17A, IL-6, IL-10, interferons (IFN)-γ, IL-12p70, CXCL8 (IL-8), and free active transforming growth factor (TGF)-β1. Following the kit instructions, data were collected by flow cytometry and analysis was performed using the LEGENDplex™ Data Analysis Software version 8.0 (Biolegend).

### 4.10. Experimental Plasmids and Transfection

The pLKO.1 vector was used to knockdown SPARC, ITGA5, and ITGB1. Stable SPARC-knockdown HL60 cells were selected using 4 μg/mL puromycin (InvivoGen. San Diego, CA, USA) for 3 days, and stable ITGA5- or ITGB1-knockdown HCT116 cells were selected using 3 μg/mL puromycin for 3 days. Protein expression was evaluated by Western blotting. SPARC was cloned into the pCMV-3 × Flag-14 vector for transient transfection of HEK293T cells. Oligonucleotide sequences used for plasmid construction are described in Appendix A.

### 4.11. Quantitative Real-Time PCR

Total RNA was extracted using the EASYspin Plus RNA Mini Kit (Aidlab. Beijing, China) and reverse-transcribed into single-stranded cDNA using the HiFiScript cDNA Synthesis Kit (CWBIO, Taizhou, China). Quantitative real-time polymerase chain reaction (RT-qPCR) was performed with SYBR Green qPCR Master Mix (Promega). Data were analyzed using LightCycler96 (Roche, Mannheim, Germany) and LightCycler96 SW 1.1 software version 1.1. Expression levels were normalized to GAPDH in each sample and subsequently standardized as fold change. Primers used for RT-qPCR are provided in Appendix A.

### 4.12. Western Blotting

Proteins were denatured in SDS loading buffer by boiling for 10 min, separated by 7.5–12.5% SDS-PAGE, and transferred to PVDF membrane (Bio-Rad, Hercules, CA, USA). Membranes were washed, blocked, and incubated with primary antibodies at 4 °C overnight. The following day, membranes were washed, incubated with horseradish peroxidase-conjugated secondary antibodies, and protein signals were detected by ECL (Merck Millipore, Billerica, MA, USA). Ponceau red was used to stain proteins in the PVDF membrane before blocking. Protein quantification was performed using ImageJ version 1.53r to analyze the intensity of protein bands or total protein. Relative expression of proteins was calculated by normalization with internal control (housekeeping protein expression (for cells) or total amount of protein (for plasma)). The antibodies used in this study are described in Appendix A. 

### 4.13. Isolation and Quantitation of NETs

NETs’ isolation was performed as previously described [22]. HL60 cells were induced using 1.25% DMSO for 4 days, and then stimulated with 20 nM phorbol 12-myristate 13-acetate (PMA) for 4 h. The supernatant was carefully aspirated and washed twice with PBS to eliminate residual PMA and non-NETs-related material. NETs were collected using serum-free DMEM, centrifuged to remove cellular debris, and stored at −80 °C until further use.

To assess the NETs formation ability, dHL60 cells from each group were seeded in 96-well plates at a concentration of 5 × 10^4^/mL. NETs were digested with DNase I (100 U/mL, Sigma-Aldrich) at 37 °C for 30 min and then stained with SYTOX Orange DNA dye (1:10,000, Invitrogen, Burlington, ONT, Canada) prior to quantitation by fluorescence measurement at 540 nm/570 nm excitation/emission.

### 4.14. Reactive Oxygen Species (ROS) Quantitation

Differentiation of sh-NC HL60 and sh-SPARC HL60 cells was induced for 4 days using RPMI-1640 containing 1.25% DMSO. Cells were incubated at 37 °C for 20 min with DCFH-DA (Beyotime, Shanghai, China) diluted at 1:1000 in serum-free medium to a final concentration of 1 μmol/L. Flow cytometry (FACSVerse) was used to detect intracellular ROS. Mean fluorescence intensity of intracellular ROS was calculated by FlowJo (Version 10.6.2).

For the PMA stimulation group, differentiation of sh-NC HL60 and sh-SPARC HL60 cells was induced for 4 days using RPMI-1640 containing 1.25% DMSO. On the fifth day, cells were stimulated with 20 nM PMA for 4 h. Cells were incubated at 37 °C for 20 min with DCFH-DA (Beyotime) diluted at 1:1000 in serum-free medium to a final concentration of 1 μmol/L. A fluorescence microplate reader (BioTEK, Santa Clara, CA, USA) was used to quantitate ROS at 488 nm/525 nm excitation/emission. 

### 4.15. Immunoprecipitation (IP)

IP experiments were conducted at 4 °C and all buffers were supplemented with protease inhibitor cocktail (MCE, Monmouth Junction, NJ, USA). For IP-MS/MS, HEK293T cells were transfected with the SPARC-3 × Flag plasmid or vector for 48 h. Cell lysate (1 mg) was incubated with anti-Flag beads (Sigma-Aldrich) at 4 °C overnight. After washing with TBS buffer, the beads were incubated with 3 × Flag peptide (300 ng/μL) at 4 °C for 2 h, and then centrifuged at 14,000 rpm for 5 min to collect the supernatant. The supernatant was boiled in SDS loading buffer and the proteins were resolved by SDS-PAGE (10%). The gel was stained with Coomassie Brilliant Blue and each lane was cut into four horizontal slices. The gel slices were processed for tandem mass spectrometry using in-gel dehydration, alkylation, trypsin digestion, and extraction. Mass spectrometry data were collected using an Orbitrap Fusion Lumos Tribrid mass spectrometer as described above.

For endogenous co-IP experiments, 1 mg cell lysate was incubated with 1 μg rabbit anti-human SPARC antibody (Abcam, Boston, MA, USA) and 1 μg rabbit anti-human IgG antibody (CST, Danvers, MA, USA) at 4 °C overnight, and subsequently immunoprecipitated with protein A/G PLUS-agarose (Santa Cruz, Dallas, TX, USA). After washing with TBS buffer, the beads were boiled in SDS loading buffer and centrifuged at 14,000 rpm for 5 min. Proteins were resolved by SDS-PAGE (10%) and subjected to Western blotting as described above.

### 4.16. SILAC-Coupled Co-Immunoprecipitation

Culture medium for stable isotopic metabolic labeling (SILAC) was prepared as previously described [69]. SILAC-1640 (Thermo Scientific) was supplemented with 10% FBS (Hyclone). ‘Heavy’ labeling contained ^13^C_6_-L-Lysine-2HCl (100 mg/L, Thermo Scientific) and ^13^C_6_-L-Arginine-HCl (50 mg/L, Cambridge Isotope Laboratories, Andover, MA, USA). HL60 cells were cultured in SILAC medium for at least 5 passages to ensure the full incorporation of SILAC amino acids into the proteins (greater than 95% incorporation). Labeled HL60 cells were induced using 1.25% DMSO for 4 days and then stimulated with 20 nM PMA for 4 h. NETs were collected using serum-free DMEM and subsequently centrifuged to remove cellular debris. HCT116 cells were incubated with labeled NETs for 24 h at 37 °C.

Cell lysate (1 mg) was incubated with 1 μg rabbit anti-human ITGB1 antibody (Abcam) and 1 μg rabbit anti-human IgG antibody (CST) at 4 °C overnight, and then immunoprecipitated with protein A/G PLUS-agarose (Santa Cruz). After washing with TBS buffer, the beads were boiled in SDS loading buffer and centrifuged at 14,000 rpm for 5 min. Proteins were resolved by SDS-PAGE (10%), the gel was stained with Coomassie Brilliant Blue, and each lane was cut into four horizontal slices. The gel slices were processed for tandem mass spectrometry using in-gel dehydration, alkylation, trypsin digestion, and extraction. Mass spectrometry data were collected using an LC-MS/MS on an Orbitrap Fusion Lumos Tribrid mass spectrometer as described above. The search included carbamidomethyl (C) as a static modification, acetylation of protein N-termini, and oxidation of methionine and heavy isotope labeling (Arg + 6, Lys + 6) as dynamic modifications.

### 4.17. Cell Proliferation Assay

The Cell Counting Kit-8 (CCK8, Beyotime) was used according to the manufacturer’s instructions to evaluate the effect of NETs on cell proliferation. A total of 2 × 10^3^ HCT116, SW620, or LoVo cells were seeded on 96-well plates and allowed to adhere overnight. Cells were incubated with NETs in the presence or absence of DNase I (100 U/mL). Proliferation was measured every 24 h for a period of 96 h. Cells were incubated with CCK-8 reagent (1:10) for 1 h prior to detection at each time point. A microplate reader was used to measure the absorbance at 450 nm.

The Cell Counting Kit-8 (CCK8, Beyotime) was used to evaluate the effect of ATN-161 on HCT116. A total of 4 × 10^3^ HCT116 were seeded on 96-well plates and allowed to adhere overnight. Cells were incubated with different concentrations of ATN-161. Proliferation was measured every 24 h for a period of 72 h. Cells were incubated with CCK-8 reagent (1:10) for 1 h prior to detection at each time point. A microplate reader was used to measure the absorbance at 450 nm.

For the colony formation experiment, 1 × 10^3^ HCT116, SW620, or LoVo cells were seeded on 6-well plates. The control group was cultured in DMEM supplemented with 5% FBS; the experimental group was cultured in DMEM supplemented with 5% FBS and containing NETs; and the DNase Ι group was cultured in DMEM supplemented with 5% FBS and containing NETs and 200 U DNase Ι. After two weeks, when colonies were visible, the culture medium was discarded and replaced with 2 mL/well 4% PFA (Solarbio) for 30 min at room temperature. Subsequently, colonies were stained with 0.1% crystal violet (Beyotime) for 3 min and then washed with PBS prior to taking images.

### 4.18. Cell Migration and Invasion Assays

To assess the effect of NETs on CRC cell migration, 5 × 10^4^/500 μL HCT116, SW620, or LoVo cells in serum-free DMEM or serum-free DMEM containing NETs were seeded onto a transwell chamber (12-well transwell chamber, 8-μm pore size, Corning, Kennebunk, ME, USA). A 300-μL aliquot of DMEM supplemented with 10% FBS was added to the lower chamber, and 200 U DNase Ι was added to the digested group.

For the invasion experiment, 5 × 10^4^/500 μL HCT116, SW620, or LoVo cells in serum-free DMEM or serum-free DMEM containing NETs were seeded onto 8-μm-pore transwell chambers with Matrigel (Corning). After a 48-h incubation, the medium was aspirated from the upper chamber and the unpenetrated cells were wiped off the membrane using a cotton swab. Cells were fixed with 4% PFA (Solarbio) and stained with crystal violet. The number of cells in six fields was counted under a microscope (Olympus, Tokyo, Japan), and the number of cells per field was calculated by ImageJ (version 1.53r).

### 4.19. Wound Healing Assay

HCT116, SW620, or LoVo cells were seeded onto 6-well plates and allowed to reach 90% confluence. Three scratches per well were made using the tip of a micropipette, after which the wells were gently washed three times with PBS to remove cell debris. Serum-free DMEM was added to the control group; serum-free DMEM containing NETs was added to the experimental group; and serum-free DMEM containing NETs and 200 U DNase Ι was added to the DNase Ι group. Cells were incubated for 0, 24, 48, and 72 h, after which representative fields were photographed. The area of each scratch was measured by ImageJ (version 1.53r), and the healing rate was calculated to reflect cell migration ability.

### 4.20. Cell Adhesion Analysis

A total of 1 × 10^6^ dHL60 cells were seeded onto 12-well plates and either left intact or stimulated with PMA (20 nM) in the presence (to form NETs) or absence of DNase I (100 U/mL) for 4 h, after which 1 × 10^5^ DiO (Beyotime)-labeled HCT116 cells were added and allowed to incubate for 30 min. Cells were subsequently washed three times with PBS and fixed with 4% PFA (Solarbio). The adhered DiO-labeled cells were directly quantitated by fluorescence microscopy (Dragonfly, Andor, Belfast, UK) of five random fields.

### 4.21. Immunohistochemistry (IHC) and Immunofluorescence (IF) Assays

After rehydration and microwave-assisted antigen retrieval, tissue slices were incubated with primary antibodies at 4 °C overnight, followed by secondary antibodies at 37 °C for 30 min. Staining was performed with 3,3-diaminobenzidine tetrahydrochloride and counterstaining was performed with Mayer’s hematoxylin. The intensity of Ly6G was classified as follows: 0, no staining; 1, weak reactivity; 2, moderate reactivity; 3, strong reactivity; 4, very strong reactivity. Images were obtained using an Olympus VS200 and analyzed by the Olympus OlyVIA software version 3.3.

Immunofluorescence staining of paraffin-embedded sections was performed similarly. After rehydration, microwave-assisted antigen retrieval, and removal of auto-fluorescence, sections were incubated with primary antibodies at 4 °C overnight, followed by fluorescence-conjugated secondary antibodies at 37 °C for 30 min. DAPI was used to stain the nuclei. Images were obtained using an Axio Scan Z1 (Zeiss, Oberkochen, Germany) and analyzed by the ZEN software (version 3.4). The antibodies used in this study are described in Appendix A.

### 4.22. Statistics Analysis

Statistical analysis was performed using GraphPad Prism version 8.0, with the exception of proteomics analysis for which R packages were used. Results are expressed as the mean ± SEM. Comparisons were performed using an unpaired Student’s *t*-test (two groups). Welch correction was applied in cases where distribution variances determined group dependence. *p* < 0.05 was considered statistically significant. Non-significant (ns) *p* > 0.05, * *p* < 0.05, ** *p* < 0.01, *** *p* < 0.001, **** *p* < 0.0001. All functional experiments were repeated independently at least three times.

## Figures and Tables

**Figure 1 ijms-24-16001-f001:**
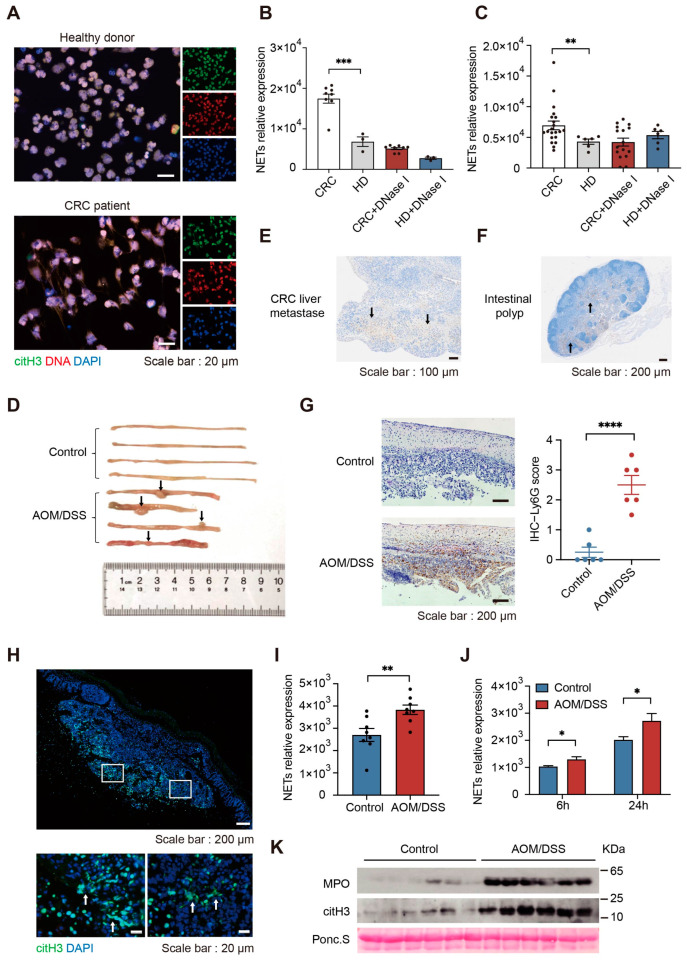
NETs formation is increased in both CRC patients and CRC mice. (**A**) Representative immunofluorescence images of neutrophil extracellular trap (NET) formation in colorectal cancer (CRC) patients and healthy donors. NETs were stained for extracellular DNA (red), citrullinated modification of histone 3 (citH3) (green), and nuclei (blue). Scale bar, 20 μm. (**B**) Fluorescence quantitation of NETs DNA released by phorbol 12-myristate 13-acetate (PMA)-stimulated neutrophils from CRC patients (*n* = 8) and healthy donors (*n* = 3). NETs were stained for extracellular DNA using SYTOX Orange, and fluorescence intensity was detected at 540 nm/570 nm (mean ± SEM, Student’s *t*-test). (**C**) Fluorescence quantitation of NETs DNA released by differentiated HL60 (dHL60) cells after a 4 h incubation in plasma from CRC patients (*n* = 21) and healthy donors (*n* = 6) (mean ± SEM, Welch’s *t*-test). (**D**) Representative images of azoxymethane (AOM)/dextran sulfate sodium (DSS)-induced CRC mouse intestines. Black arrows indicate intestinal polyps. (**E**) Representative images of Ly6G immunohistochemistry staining of liver metastases in CRC mice. Black arrows indicate typical neutrophil infiltration areas. Scale bar, 100 μm. (**F**) Representative images of lymphocyte antigen 6 complex locus G6D (Ly6G) immunohistochemistry staining of intestinal polyps in CRC mice. Black arrows indicate typical neutrophil infiltration areas. Scale bar, 200 μm. (**G**) Representative images of Ly6G immunohistochemistry staining of intestines in CRC mice (Left). Scale bar, 200 μm. Immunohistochemical scoring statistics (Right) (*n* = 6 per group; mean ± SEM, Student’s *t*-test). (**H**) Representative images of citH3 immunofluorescence staining of intestinal polyps in CRC mice. White arrows indicate NETs. Scale bars, 200 μm and 20 μm. (**I**) Fluorescence quantitation of NETs DNA released by neutrophils from CRC mice and healthy mice (*n* = 8; mean ± SEM, Student’s *t*-test). (**J**) Fluorescence quantitation of NETs DNA released by dHL60 cells after a 6 h (*n* = 8) and 24 h (*n* = 6) incubation in plasma from CRC mice and healthy mice (mean ± SEM; 6 h, Welch’s *t*-test; 24 h, Student’s *t*-test). (**K**) Western blotting analysis of myeloperoxidase (MPO) and citH3 expression in plasma from CRC mice and control. The expression levels of MPO and citH3 in plasma are increased in CRC mice. * *p* < 0.05, ** *p* < 0.01, *** *p* < 0.001, **** *p* < 0.0001.

**Figure 2 ijms-24-16001-f002:**
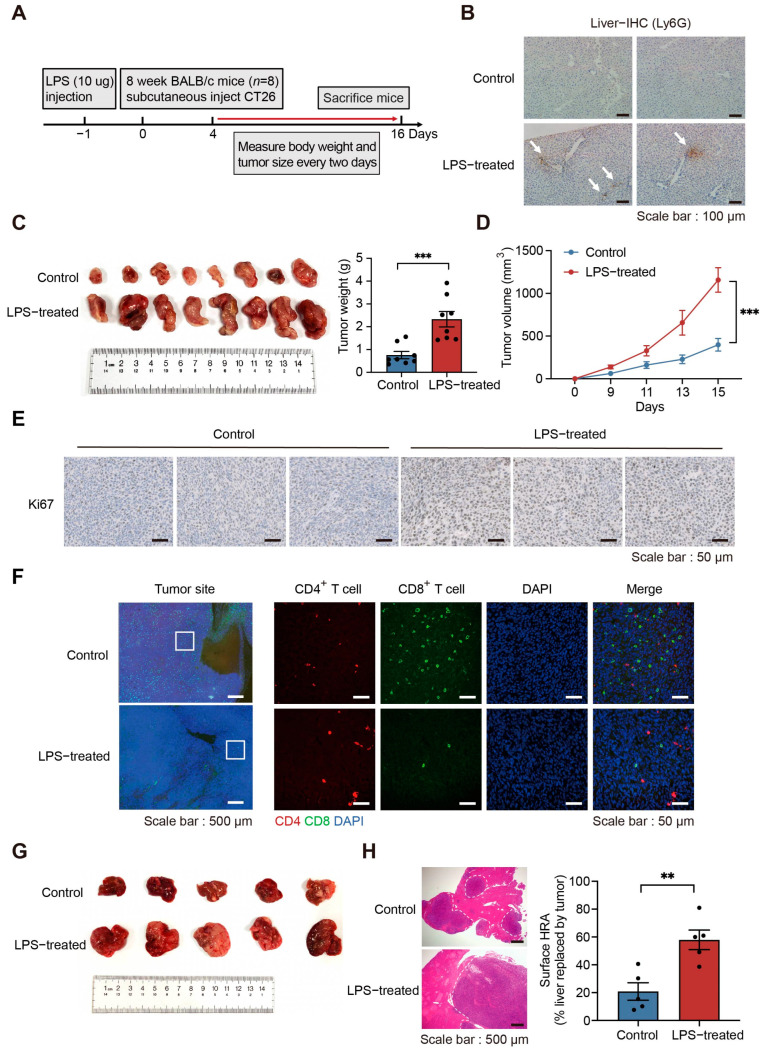
NETs promote tumor growth and metastasis in vivo. (**A**) Schematic diagram of the construction of the subcutaneous tumor-bearing NETs murine model. (**B**) Representative images of Ly6G immunohistochemistry staining of the liver from lipopolysaccharide (LPS)-treated mice. White arrows indicate typical neutrophil infiltration areas. Scale bar, 100 μm. (**C**) Photographs of mouse tumors at the end of the experiment. The subcutaneous tumors grow faster in LPS-treated mice (*n* = 8 per group; mean ± SEM, Student’s *t*-test). (**D**) Tumor volumes were measured at different time points after inoculation (*n* = 8 per group; mean ± SEM, Student’s *t*-test). (**E**) Representative images of Ki67 immunohistochemistry staining of tumors from LPS-treated mice. Scale bar, 50 μm. (**F**) Reduced infiltration of CD8^+^ T cells at the tumor sites in LPS-treated mice. Representative fluorescence images are shown. Scale bars, 500 μm (Left) and 50 μm (Right). (**G**) Photographs of mouse livers at the end of the experiment. Enhanced liver metastasis is seen in LPS-treated mice. (**H**) Representative images of hematoxylin-eosin (HE) staining of liver metastases in LPS-treated mice (Left). Scale bar, 500 μm. Statistics of liver replacement area (Right; *n* = 5 per group; mean ± SEM, Student’s *t*-test). ** *p* < 0.01, *** *p* < 0.001.

**Figure 3 ijms-24-16001-f003:**
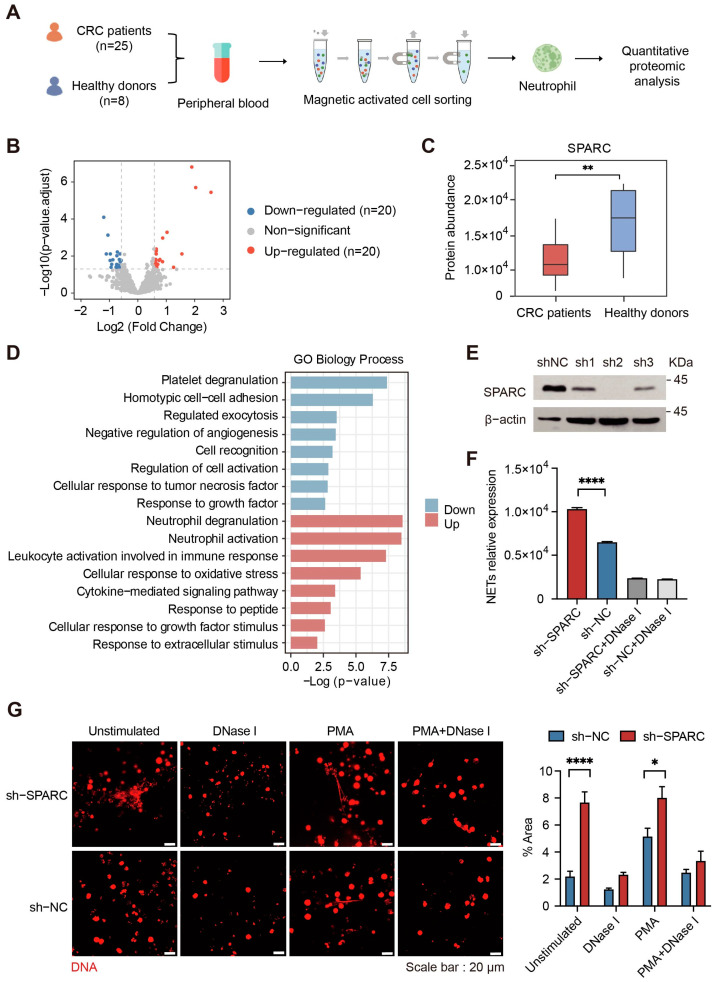
Downregulation of SPARC promotes the formation of NETs. (**A**) Schematic diagram of the quantitative proteomic analysis of neutrophil derived from CRC patients and healthy donors. (**B**) Volcano plots for the neutrophil proteomes showing significantly changed proteins with an adjusted *p* value < 0.05 and an absolute value of Log2 (fold change) > 0.5 (calculated as stated in the methods). (**C**) The abundance of secreted protein acidic and rich in cysteine (SPARC) in neutrophils from healthy donors and CRC patients. SPARC is significantly downregulated in CRC patients (adjusted *p*-value was calculated by DESeq2 (version 1.24.0), Wald-test). (**D**) Biological function annotation enrichment analysis of significantly changed proteins in CRC neutrophils. (**E**) Western blotting analysis of SPARC expression in sh-NC and SPARC-knockdown HL60 cells. (**F**) Fluorescence quantitation of NETs DNA released in SPARC-knockdown and sh-NC HL60 cells. Deoxyribonuclease I (DNase I) was added 30 min prior to quantitation (*n* = 6 per group; mean ± SEM, Student’s *t*-test). (**G**) Representative fluorescence images of NETs formation in SPARC-knockdown (sh-SPARC) and negative control (sh-NC) HL60 cells. PMA treatment condition: 20 nM PMA stimulation for 4 h. DNase Ι was added 30 min prior to fluorescence observation. NETs were stained for extracellular DNA (red). Scale bar, 20 μm. Fluorescence values were analyzed using ImageJ (version 1.53r) (*n* = 8 per group; mean ± SEM, Student’s *t*-test). * *p* < 0.05, ** *p* < 0.01, **** *p* < 0.0001.

**Figure 4 ijms-24-16001-f004:**
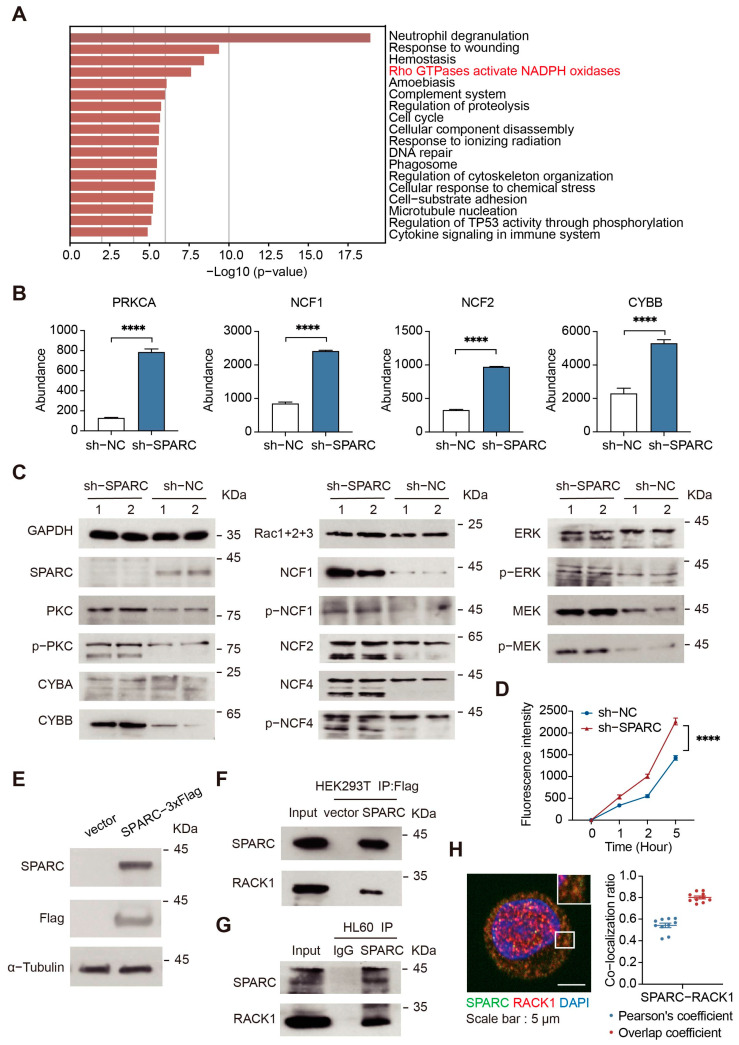
SPARC affects NETs formation by regulating the NADPH oxidase pathway. (**A**) Biological function annotation enrichment analysis of significantly changed proteins in SPARC-knockdown HL60 cells. (**B**) The abundance of protein kinase C (PRKCA), neutrophil cytosol factor 1 (NCF1), neutrophil cytosol factor 2 (NCF2), and cytochrome b-245 beta chain (CYBB). After knockdown of SPARC, NADPH oxidase pathway-related proteins are significantly upregulated (*n* = 3 per group; mean ± SEM, adjusted *p*-value was calculated by DESeq2 (version 1.24.0), Wald-test). (**C**) NADPH oxidase-related pathway protein expression and phosphorylation levels in HL60 cells were assessed by immunoblotting. After knockdown of SPARC, the protein expression and phosphorylation levels of NADPH oxidase-related pathway proteins are upregulated. (**D**) After stimulation of sh-NC and SPARC-knockdown dHL60 cells with 20 nM PMA, the reactive oxygen species (ROS) detection probe (DCFH-DA) was loaded and the fluorescence intensity was measured at 1 h, 2 h, and 5 h (*n* = 12 per group; mean ± SEM, Student’s *t*-test). (**E**) Western blotting analysis of SPARC-3 × Flag expression in HEK293T cells. The SPARC-3 × Flag plasmid was constructed for the subsequent identification of SPARC interacting proteins by mass spectrometry. (**F**) Interaction between exogenous SPARC and the receptor of activated protein C kinase 1 (RACK1) in HEK293T cells was detected by Co-IP using anti-Flag beads. (**G**) Interaction between endogenous SPARC and RACK1 in HL60 cells was detected by Co-IP using IgG or SPARC antibodies. (**H**) Representative fluorescence images of the colocalization of SPARC (green) and RACK1 (red). Scale bar, 5 μm. Colocalization ratios were analyzed using ImageJ (version 1.53r) (*n* = 10 per group, mean ± SEM). **** *p* < 0.0001.

**Figure 5 ijms-24-16001-f005:**
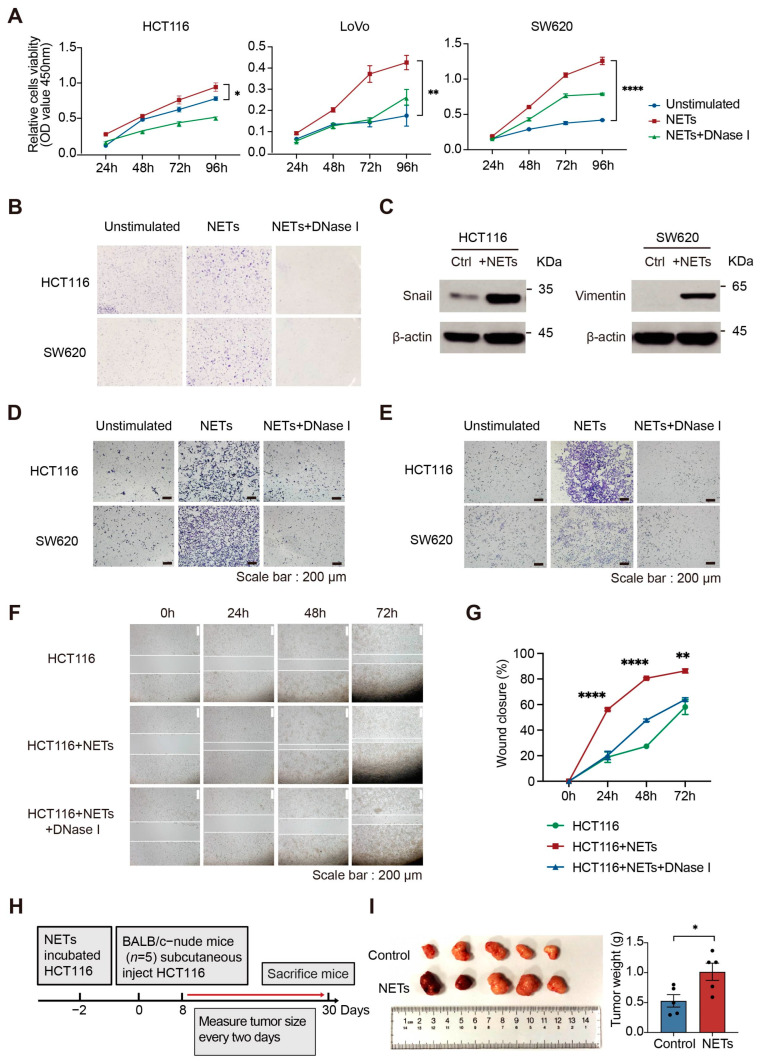
NETs capture and promote CRC cell proliferation, migration, and invasion in vitro. (**A**) NETs promote the proliferation of CRC cells in the cell counting kit 8 (CCK8) assay, which is attenuated by the addition of DNase I (*n* = 4 per group; mean ± SEM, HCT116 and LoVo: Student’s *t*-test, SW620: Welch’s *t*-test). (**B**) CRC cells incubated with NETs show enhanced clonogenicity. Cells were stained with crystal violet. (**C**) The expression levels of snail, vimentin, and Ki67 are increased following incubation of CRC cells (HCT116 and SW620) with NETs. (**D**,**E**) Representative images of the transwell chamber assay. NETs promote the migration (**D**) and invasion (**E**) of CRC cells (HCT116 and SW620), which is abolished by DNase I. Scale bar, 200 μm. (**F**) The effect of NETs on the migration ability of HCT116 cells was detected using a wound healing assay; representative images are shown. Scale bar, 200 μm. (**G**) The wound healing rate of each group and their representative photographs were analyzed using ImageJ (version 1.53r) (*n* = 5 per group; mean ± SEM, 48 h: Student’s *t*-test, 24 h and 72 h: Welch’s *t*-test). (**H**) Schematic diagram of the establishment of a subcutaneous tumor model in BALB/c-nude mice. HCT116 cells were incubated with NETs for 48 h in vitro. (**I**) Photographs of mouse tumors at the end of the experiment. The subcutaneous tumors grow faster in NETs-treated mice (*n* = 5 per group; mean ± SEM, Student’s *t*-test). * *p* < 0.05, ** *p* < 0.01, **** *p* < 0.0001.

**Figure 6 ijms-24-16001-f006:**
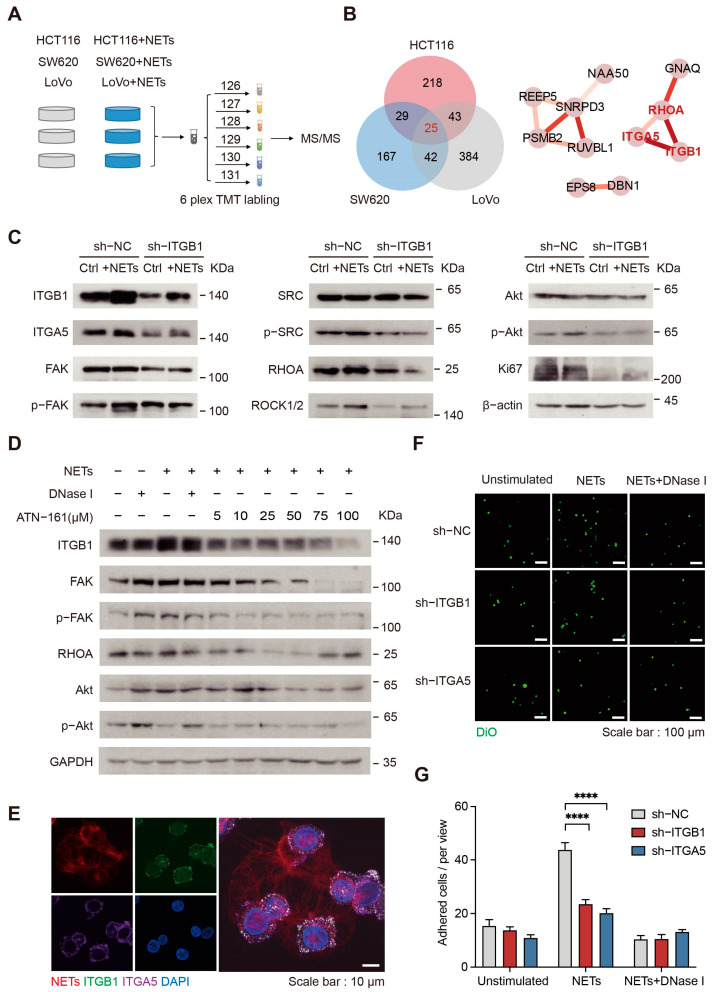
NETs activate integrin pathways in CRC cells. (**A**) Schematic of the experimental design and quantitative proteomics workflow. CRC cell lines, HCT116, SW620, and LoVo, were incubated with NETs for 48 h. (**B**) The intersection of significantly changed proteins in HCT116, SW620, and LoVo cells (Left). The interaction of significantly changed proteins: the color of the lines represents the degree of interaction (Right). (**C**) Western blotting analysis of integrin α5β1 signaling pathway proteins in HCT116 cells with NETs’ incubation. Tumor cell integrin signaling pathways are activated following incubation with NETs, which is attenuated after ITGB knockdown. (**D**) Western blotting analysis of integrin α5β1 signaling pathway proteins in HCT116 cells under different conditions. Tumor cell integrin signaling pathways are activated following incubation with NETs, which is attenuated after addition of the integrin α5β1 inhibitor ATN-161(Ac-PHSCN-NH2). (**E**) Representative immunofluorescence images of NETs-tumor cell interactions. Scale bar, 10 μm. (**F**) Representative immunofluorescence images of NETs-captured tumor cells after knockdown of ITGA5 or ITGB1. Scale bar, 100 μm. (**G**) Quantitation of CRC cells captured by NETs in vitro. CRC cell number was calculated using ImageJ (version 1.53r) (*n* = 8 per group; mean ± SEM, Student’s *t*-test). **** *p* < 0.0001.

**Figure 7 ijms-24-16001-f007:**
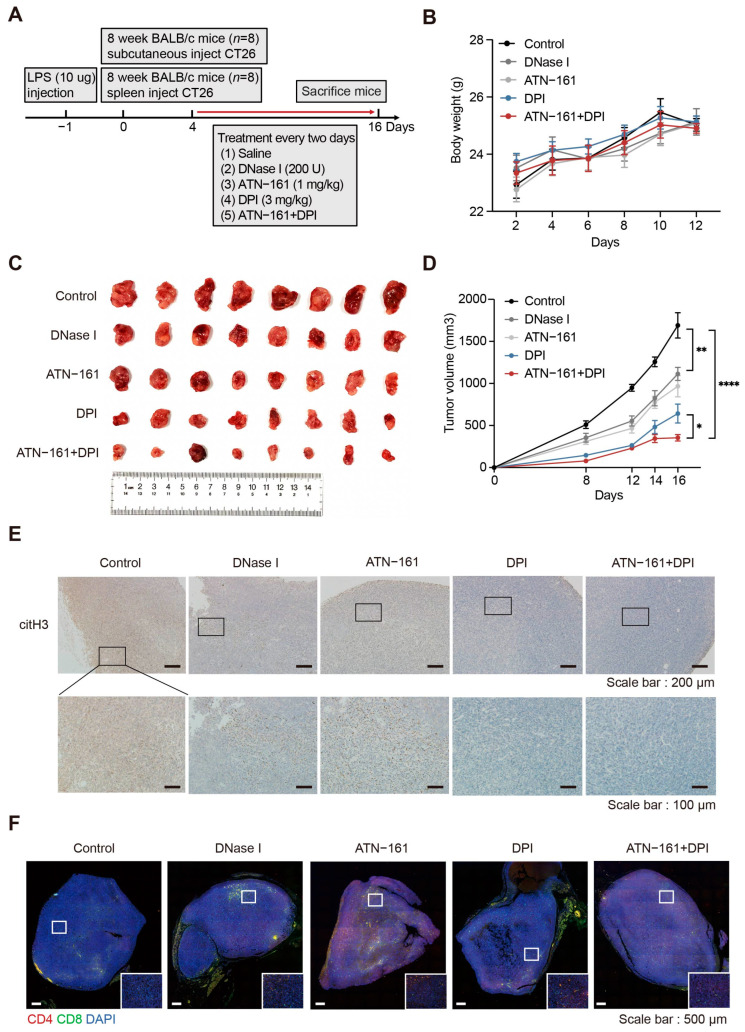
Simultaneous targeting of NADPH oxidase and integrin α5β1 inhibits tumor growth. (**A**) Experimental design of drug treatment in the subcutaneous tumor and liver metastasis murine models. (**B**) Bodyweight curves for the different treatment groups (*n* = 8 per group; mean ± SEM, Student’s *t*-test). (**C**) Photograph of mouse tumors at the end of the experiment. (**D**) Tumor growth curves for the different treatment groups (*n* = 8 per group; mean ± SEM, Student’s *t*-test). (**E**) Representative images of citH3 immunohistochemistry staining of the tumor site in CRC mice. Scale bars, 200 μm and 100 μm. (**F**) Drug therapy increases T cell infiltration at the tumor site. Representative fluorescence images are shown. Scale bar, 500 μm. * *p* < 0.05, ** *p* < 0.01, **** *p* < 0.0001.

**Figure 8 ijms-24-16001-f008:**
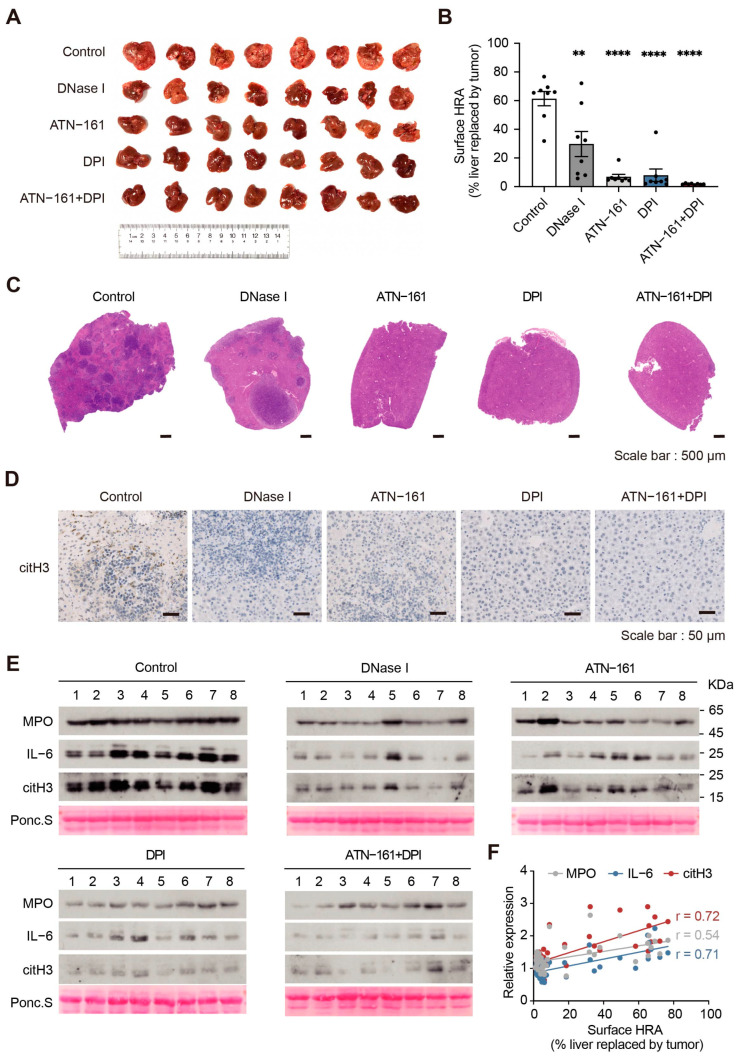
Simultaneous targeting of NADPH oxidase and integrin α5β1 inhibits tumor metastasis. (**A**) Photograph of mouse livers at the end of the experiment. (**B**) Statistics of the liver replacement area (*n* = 8 per group; mean ± SEM, Student’s *t*-test). (**C**) Representative images of HE staining of liver metastasis in CRC mice. Scale bar, 500 μm. (**D**) Representative images of citH3 immunohistochemistry staining of liver metastasis in CRC mice. Scale bar, 50 μm. (**E**) The expression levels of MPO, IL-6, and citH3 in plasma from mice in the different drug treatment groups were assessed by immunoblotting. (**F**) The positive correlation between the liver area replaced by tumor and the expression levels of NETs-related proteins (MPO, IL-6, and citH3) in plasma are shown in the scatterplot. (MPO, r = 0.54, *p* = 0.0003; IL-6, r = 0.71, *p* < 0.0001; citH3, r = 0.72, *p* < 0.0001; *n* = 40). ** *p* < 0.01, **** *p* < 0.0001.

## Data Availability

Quantitative proteomic datasets generated and/or analyzed in this study have been deposited to the ProteomeXchange Consortium (http://proteomecentral.proteomexchange.org (accessed on 27 October 2023)) via the iProX partner repository with the dataset identifier PXD046490. The significantly changed proteins and SILAC-modified peptides are included as Appendix A.

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
