# Peer review of "Targeting NADPH Oxidase and Integrin α5β1 to Inhibit Neutrophil Extracellular Traps-Mediated Metastasis in Colorectal Cancer"

_ijms, 2023, doi:10.3390/ijms242116001_

Round 1

Reviewer 1 Report

Comments and Suggestions for Authors

This research titled "Targeting NADPH oxidase and integrin α5β1 to inhibit neutrophil extracellular traps-mediated metastasis in colorectal cancer" focuses on understanding the role of Neutrophil Extracellular Traps (NETs) in metastasis, which is a significant cause of mortality in colorectal cancer (CRC). The authors have identified that an increase in NET formation is a key factor in metastasis. The study uses quantitative proteomics analysis of clinical samples and cell lines to show that decreased SPARC (Secreted Protein Acidic and Rich in Cysteine) results in increased NET formation. They also identified integrin α5β1 as a crucial protein involved in interacting with NETs and tumor cells. The study suggests a combination therapy using an NADPH oxidase inhibitor (diphenyleneiodonium chloride, DPI) and an integrin α5β1 inhibitor (ATN-161) to suppress tumor progression. While the authors have made commendable efforts to add antibody information and full-length SDS-PAGE membranes in the supplement section, there are still significant issues in the western blotting experimental section and its representation that need to be addressed to make the scientific observations sound credible. However, there are some major things to be done in the western blotting experimental section and its representation to make the scientific observations sound credible. 

  • Western blotting does not contain the comparative profiling, number of biological replicates, and normalized protein expression with their respective housekeeping gene expression. In the representation on page number 17 Figure 8 E, the authors have described the expression levels of 429 MPO, IL-6, and citH3 in plasma from mice in the different drug treatment groups on separate gels. 
  • These are great for quantification, but, the authors are requested to depict the comparative expression levels of 429 MPO, IL-6, and citH3 groups adjacent to each other.  
  • The materials section does not describe the quantification methodology and the method involved.
  • The Graphical abstract on page 27 contains a "FIG DRAW" water marker, remove it in the final version.  

The authors are requested to make the above-mentioned changes. Overall the research provides valuable insight into the mechanistic link between NET and tumor progression and proposes a potential therapeutic strategy for combating NET-mediated metastasis in CRC. Best

Comments on the Quality of English Language

No significant changes are required to the technical and grammatical sections of the manuscript. 

Reviewer 2 Report

Comments and Suggestions for Authors

The study titled " Targeting NADPH oxidase and integrin α5β1 to inhibit neutrophil extracellular traps-mediated metastasis in colorectal cancer" describes the mechanistic connection between NETs and tumor progression. The manuscript is important and needs to improve the presentation of this manuscript. Here are the specific comments.

-This would be beneficial if the authors rewrite the conclusion in more reader-friendly.

-Lots of western blot is not clear for which cell lines (the presentation is not clear).

-Antibodies information is not for all of what is presented.

-The presentation of the different cell lines benefits is not enough if any and is not organized.

Comments on the Quality of English Language

The language is good but the organization needs to improve.

Reviewer 3 Report

Comments and Suggestions for Authors

The provided abstract presents a study on colorectal cancer (CRC) and the role of neutrophil extracellular traps (NETs) in CRC metastasis. The study explores the underlying mechanisms and potential therapeutic strategies for inhibiting NETs-mediated metastasis in CRC. Here's a short review of the text:

Strengths:

  1. Clear Objective: The study aims to investigate the role of NETs in CRC metastasis and potential therapeutic interventions. The objective is well-defined.

  2. Detailed Mechanistic Insights: The abstract provides detailed mechanistic insights into how NETs promote tumor progression, involving NADPH oxidase, integrin α5β1, and several signaling pathways.

  3. Clinical Relevance: The study is significant due to the clinical relevance of CRC and its high mortality rate. Understanding the mechanisms behind NETs-mediated metastasis is crucial for developing effective therapies.

  4. Combination Therapy Approach: The abstract highlights the potential of combination therapy involving NADPH oxidase inhibitor DPI and integrin α5β1 inhibitor ATN-161 to suppress tumor progression.

Areas of Improvement:

  1. Clarity and Organization: While the abstract contains valuable information, it could benefit from improved organization and clarity. It feels somewhat disjointed, making it challenging for readers to follow the logical flow of the study. Separating the abstract into distinct sections with subheadings could improve clarity.

  2. More Concise Language: Some sentences are lengthy and could be condensed for greater readability. Concise language will help in conveying complex scientific information more effectively.

  3. Visual Aids: The inclusion of visual aids such as diagrams or figures might help readers better grasp the intricate molecular processes and interactions discussed in the study.

  4. Introduction and Conclusion: The abstract lacks a succinct introductory statement and a clear conclusion summarizing the key findings. These additions can provide a better context and closure to the study.

  5. Citation Format: The citations in the text should be properly formatted according to a specific citation style, such as APA or MLA, for academic and scientific rigor.

  6. Define Abbreviations: Some abbreviations (e.g., CRC) are defined, while others (e.g., DPI) are not. Consistently defining all abbreviations upon first use would enhance readability.

Overall, the abstract presents an important study but could benefit from improved organization, conciseness, and additional context to make it more accessible to a broader scientific audience.
